# Earthquake slip surfaces identified by biomarker thermal maturity within the 2011 Tohoku-Oki earthquake fault zone

Hannah S. Rabinowitz [1]*, Heather M. Savage[2], Pratigya J. Polissar[3], Christie D. Rowe [4] & James D. Kirkpatrick [4]

Extreme slip at shallow depths on subduction zone faults is a primary contributor to tsunami generation by earthquakes. Improving earthquake and tsunami risk assessment requires understanding the material and structural conditions that favor earthquake propagation to the trench. We use new biomarker thermal maturity indicators to identify seismic faults in drill core recovered from the Japan Trench subduction zone, which hosted 50 m of shallow slip during the $M_w$9.1 2011 Tohoku-Oki earthquake. Our results show that multiple faults have hosted earthquakes with displacement $\geq$ 10 m, and each could have hosted many great earthquakes, illustrating an extensive history of great earthquake seismicity that caused large shallow slip. We find that lithologic contrasts in frictional properties do not necessarily determine the likelihood of large shallow slip or seismic hazard.

[1] AAAS Science and Technology Policy Fellow at the U.S. Department of Energy, 955 L'Enfant Plaza SW, Washington, DC 20024, USA. [2] Department of Earth and Planetary Sciences, University of California, Santa Cruz, 1156 High St, Santa Cruz, CA 95064, USA. [3] Department of Ocean Sciences, University of California, Santa Cruz, 1156 High St., Santa Cruz, CA 95064, USA. [4] Department of Earth and Planetary Sciences, McGill University, 3450 University St, Montreal, QC H3A 0E8, Canada. *email: hsrabinowitz@gmail.com

During large subduction earthquakes, coseismic slip can propagate to the seafloor and increase the severity of seismic hazards such as earthquake-related tsunamis. The 11 March 2011, $M_w$9.1 Tohoku-Oki earthquake was one such event. The earthquake and accompanying tsunami led to significant damage in Japan, claimed the lives of over 15,000 people, and caused a meltdown in the Fukushima Daiichi nuclear power plant[1]. The tsunami was enhanced by an estimated ~50 m of shallow earthquake slip that propagated to the subduction trench[2–4]. On-shore tsunami deposits suggest that at least three similar tsunamis have occurred along this section of the Japan Trench with ~1000 years periodicity[5]. The conditions that allow for shallow coseismic slip are not well understood, but could depend on such parameters as interactions with the seafloor[6], frictional properties of the fault zone[7,8], and dynamic weakening of faults during rapid slip[9–11]. Improving earthquake risk assessment requires understanding the conditions that favor earthquake propagation to the trench. Here, we use indicators in the rock record to determine which faults within the Japan Trench experienced earthquake slip, in order to understand how shallow slip relates to the material properties of subducting sediments.

International Ocean Discovery Program (IODP) Expedition 375, JFAST, drilled through the Japan Trench where the maximum slip occurred during the Tohoku-Oki earthquake in order to study the physical controls on shallow seismic slip (Fig. 1)[13]. Structural and stratigraphic characterizations show that the stratigraphy in the recovered core is comprised of mudstones and pelagic clays offset by several faults[17–19]. One of these faults, located within a frictionally weak, thin pelagic clay with a penetrative scaly fabric at ~820 mbsf has been interpreted as a major structural boundary[13]. Structural features and sediment age also significantly change from one side of this fault to the other[18–20]. Because of these

observations, most of the estimated 3.2 km displacement of the subducted oceanic plate—including earthquakes—are thought to have localized on this fault[7–9,11,13,18,20,21]. However, other faults are present and form part of the subduction interface fault system, and it can be difficult to determine whether a fault has failed seismically or aseismically based on structural evidence alone.

The most robust, independent indication of seismic slip is temperature rise along the fault, because the significant temperature rise ($>\sim100$ °C) only occurs during rapid slip, i.e., earthquakes[22]. The JFAST expedition installed a temperature observatory to monitor the decay of the heat signal produced by frictional heating during the Tohoku-Oki earthquake. The measured temperature anomaly suggests a low-integrated coseismic shear stress of 0.54 MPa with a source depth of ~820–822 mbsf[23]. However, there are several closely spaced faults within the bottom 15 m of the core (~820–835 mbsf)[17–19] and due to the diffuse nature of the measured temperature anomaly[13], Fulton et al.[23] are unable to constrain which specific fault slipped during the earthquake or whether several of these faults may have slipped and contributed heat.

A different way to determine whether the faults at JFAST have experienced elevated temperatures is to look for evidence in the fault rocks themselves. Thermal alteration inside or near the fault relative to the surrounding rock provides evidence of earthquake slip. Although this approach cannot define the slip in any specific past earthquake, it places a bound on the highest temperature the fault has ever achieved during earthquake slip. Therefore, thermal alteration can determine which faults have hosted large earthquakes such as the Tohoku-Oki earthquake. Coseismic temperature rise in subduction zones has been investigated through a variety of proxies including vitrinite reflectance, pseudotachylyte, clay alteration, and changes in magnetic susceptibility[24–28]. Here, we analyze the thermal maturity of organic matter

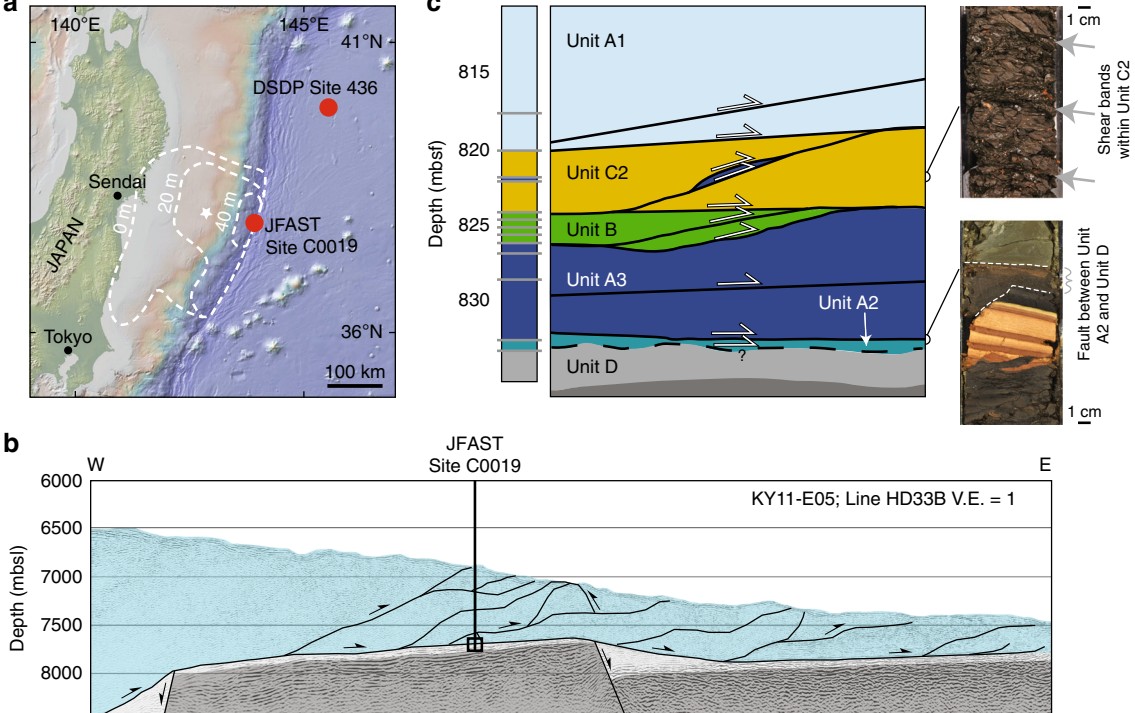

**Fig. 1 Location and structure of the JFAST core. a** Map of the Japan Trench with the JFAST and Site 436 sites identified by red dots. White dotted contours indicate regions of equal slip during the Tohoku-Oki earthquake as inferred by geophysical data[12–15]. **b** Schematic structure of the accretionary prism perpendicular to the trench recovered at JFAST traced over a seismic section[16], and **c** close-up of the schematic JFAST stratigraphy with interpreted structure[17, 18] and pictures of typical structures from regions of the core where damage has been observed.

(molecular biomarkers) in and around faults within the JFAST core. Biomarkers experience irreversible chemical alterations that permanently record the maximum temperature experienced by a fault hosted in sedimentary sequences, and have been used to detect heating signatures in ancient faults[29–31]. Recent work establishing reaction kinetics of biomarkers at nearly seismic timescales[32,33] allows for estimates of peak temperature in the JFAST faults, and high-velocity friction experiments show that biomarkers can react at earthquake timescales during shearing[34].

We use thermal alteration of coccolithophore algae-sourced long-chain unsaturated ketones (herein alkenones)[35] and plant-derived long-chain $n$-alkane distributions to identify heating anomalies. Alkenones document heating through decreasing concentrations ($C_{37}$ total)[33,36] and preferential destruction of molecules with three double bonds compared with those with two double bonds (measured by increasing $U_{37}^{k'}$ values)[33]. $n$-Alkane distributions document heating through a decreasing carbon preference index (CPI) and by the addition of a secondary peak in the carbon chain-length distribution described by a decrease in the alkane distribution index (ADI, "Methods")[33]. The alteration of each biomarker is cumulative and increased thermal maturity in a biomarker proxy represents the degree of frictional heating that has occurred on a fault integrated over its entire slip history. However, this effect is nonlinear. Earthquakes that cause the high temperature rise will have a greater influence on the cumulative thermal maturity than multiple smaller events[31]. We use kinetic parameters for these reactions[33] to infer heating during earthquake slip from reaction extent. Differences in kinetic parameters for the various reactions mean that differential alteration can be used to help constrain temperature rise during earthquake slip. Our results demonstrate that multiple faults in different lithologies have hosted earthquake slip ≥10 m within the shallow fault zone at the Japan Trench.

## Results

**Biomarker analysis.** In order to use biomarker thermal maturity to reconstruct temperature rise, it is necessary to establish the initial values of the biomarker parameters before the rocks were faulted. We do not use samples from the JFAST core for this purpose to avoid mistakenly including a thermally altered sample in the initial values. Instead, we compare JFAST samples to a reference core on the incoming plate (DSDP Site 436, Fig. 1a) that has not been tectonically deformed to determine the level of biomarker alteration. Stratigraphic units were correlated between the two cores using trace element concentrations[17], which provide a way to accurately fingerprint units with similar lithologies and ages on a sample-by-sample basis. This stratigraphy demonstrates the presence of age gaps and inversions interpreted as faults at various depths (Fig. 1c, Supplementary Note 1)[17], and is consistent with faults identified through biostratigraphic age inversions and damage structures in the JFAST core[13,18,19].

We analyzed the fractional change ($r$) of $C_{37}$ total, $U_{37}^{k'}$, CPI and ADI in JFAST samples compared with Site 436 as the ratio of the biomarker parameter measured in a JFAST sample ($H_{JFAST}$) to the initial values measured from Site 436 ($H_{initial}$): $r = H_{JFAST}/H_{initial}$ (Fig. 2). For each JFAST sample, the distribution of $r$ is calculated from the distribution of $H_{initial}$ values measured in the corresponding sedimentary unit at Site 436 (Supplementary Figs. 1–5). Faults that have experienced coseismic frictional heating are defined by values of $r$ less than one for $C_{37}$ total, CPI, and ADI and values of $r$ greater than one for $U_{37}^{k'}$.

Because each sedimentary unit has a range of $H_{initial}$ values, the $r$ values show a range as well (represented through the size of the box and whiskers for each sample). The $H_{initial}$ biomarker values are a function of various parameters such as ocean temperature, productivity, and sedimentation rate, and they vary both within and between units. Supplementary Figs. 1–5 show the initial

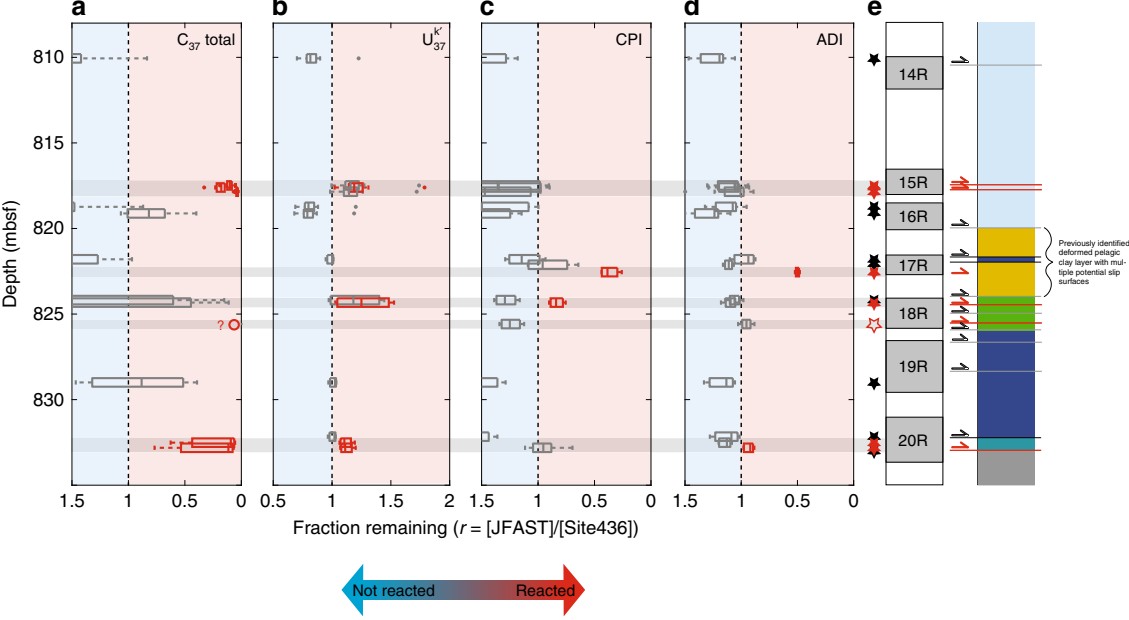

**Fig. 2 Biomarker indicators of heating in the JFAST core.** Fraction remaining (i.e., not reacted) of (**a**) $C_{37}$ total, (**b**) $U_{37}^{k'}$, (**c**) CPI, and (**d**) ADI are shown for samples in the plate boundary region. Box plots indicate the median and quartiles of the fraction remaining value (relative to the range of biomarker values measured in the corresponding sedimentary unit at Site 436) while minimum and maximum values are indicated by the whiskers, and outliers by dots. **e** Sample locations are shown as stars, colored red when biomarker anomalies indicate heating. Gray shading indicates JFAST core recovery while white gaps indicate missing section. Stratigraphy is shown with previously observed faults indicated with arrows[17–19], those with biomarker anomalies discussed in the main text in red, and those lacking an anomaly, in black. Light gray bars crossing between plots show the depths at which biomarker anomalies are observed. Sample PP948, represented by hollow symbols in **a** and **e**, had alkenone concentrations below the quantification limit and thus, the magnitude of the heating anomaly is poorly constrained.

range of each parameter measured in the Site 436 core. We can evaluate how well $H_{initial}$ values from Site 436 represent unaltered samples at JFAST by examining $r$ values from off-fault JFAST samples that are less likely to have been heated. For the two alkenone parameters ($C_{37}$ total and $U^{k'}_{37}$), all of the off-fault JFAST sediments have biomarker values that fall within the range for the equivalent sediment unit in the Site 436 core (i.e., the range of $r$ values includes one) (Supplementary Figs. 1–2). The $n$-alkane parameters suggest that the off-fault JFAST core values are either the same or higher than Site 436 samples, perhaps due to the more south and westward position of the JFAST location with greater contributions of organic material from land[37] (Supplementary Figs. 4–5). If we used off-fault samples from JFAST sediments of equivalent units, we might identify a few more samples as heated in their $n$-alkane parameters, but in most cases, these faults are already identified as heated from the alkenone measurements and thus would not change our overall findings. We take a conservative approach and only consider a sample reacted if the entire range of $r$ values is less than one for a given biomarker parameter (or greater than one for $U^{k'}_{37}$). This means that the sample is different from the entire range of values for a given biomarker found in Site 436. Because of this approach, we are unlikely to erroneously identify a sample as thermally mature (false positive), but we may underestimate the thermal maturity of some samples (false negative).

Our biomarker analysis indicates that at least eight of the faults we sampled have experienced shear heating at seismic slip rates (red symbols in Fig. 2). Gray arrows in Fig. 2 represent faults that did not show a thermal anomaly, although we only sampled close enough to three of those faults that a heating signature could have been detected but was not. Some of the seismic faults only show a heating anomaly in the alkenone proxies, while others show alteration in both the alkenone and $n$-alkane proxies. These differences are due to the faster reaction rates of alkenones compared with $n$-alkanes (Supplementary Table 1), and alteration of both proxies on some faults would be caused by earthquakes that generated higher temperatures and/or longer durations of heating[33]. The pelagic clay layer has initial alkenone concentrations below the detection limit, so the temperature anomaly there is based solely on the $n$-alkanes. Of the two pelagic clay samples that we analyzed, only one exhibited biomarker anomalies (PP829, 822.55 mbsf), implying that seismic slip in this weak layer took place on select localized features[18,27]. We note that the incomplete core recovery (Fig. 2) means that there could be more seismic faults that were not sampled. In summary, our biomarker analysis shows that selected faults within a 15-m thick zone experienced differential heating from seismic slip, and this slip occurred within many of the stratigraphic units found in the core.

**Forward modeling of earthquake slip**. Biomarker thermal maturity anomalies indicate heating of faults relative to the protolith and surrounding rock caused by earthquake slip. However, the anomalies are integrative and it is not possible to ascribe anomalies to a specific earthquake or to a certain number of earthquakes on that fault. Nevertheless, we can place important constraints on the earthquake conditions that could or could not produce the thermal maturity anomalies (Fig. 3e). Furthermore, we can also test whether the anomalies allow for a seismic rupture with a specific set of earthquake conditions, such as those of the 2011 Tohoku-Oki earthquake where shear stress, displacement, and duration are constrained.

Here we couple a model for heat generation and dissipation from earthquake slip[22,38,39] with the kinetics for the reaction of biomarker thermal maturity parameters derived from laboratory experiments[33] to forward model the effect of different earthquake

conditions on those maturity parameters. We ask two specific questions with our models. First, do the maturity anomalies on each fault allow for the Tohoku-Oki earthquake on that fault? Second, what is the smallest earthquake size that can generate the thermal maturity anomaly we observe? For this second question we limit the number of earthquakes in the model to produce total offset less than or equal to the estimated 3.2 km of total displacement on the décollement[13].

**Modeling biomarker reaction kinetics**. The reaction kinetics for $C_{37}$ total, $U^{k'}_{37}$, CPI, and ADI were measured from hydrous pyrolysis experiments at minutes to hours timescales and from 100 to 385 °C[33]. Some of the modeled temperatures are above our laboratory temperature range and we extrapolate the kinetics (with uncertainty) to those temperatures. Rabinowitz et al.[33] found that rates of alkenone destruction and $U^{k'}_{37}$ alteration at 100 °C were much lower than expected from experiments at 120 °C or above. We therefore limit biomarker reaction in the model to 120 °C or above using the temperature-dependent kinetics determined at those higher temperatures. Our modeling results are not sensitive to this choice as all possible solutions require temperature rise greatly in excess of this cutoff temperature, and there are not solutions that would be allowed by extending reaction to lower temperatures.

**Calculating coseismic heating**. The heat generated during an earthquake is the product of the shear resistance and the total slip of the earthquake. Temperature rise is a function of the rate of heat generation (slip velocity), the volume in which heating occurs (slipping layer thickness), and the rate of diffusive heat loss (thermal diffusivity of the rock)[22] ("Methods"). For the Tohoku-Oki earthquake, the shear resistance, total slip, and slip velocity are known through seismic, geodetic, and geophysical data[4,23,40] (Supplementary Table 1). We constrain maximum fault thicknesses through core image analysis of structures in the vicinity of biomarker anomalies (Supplementary Fig. 6, Supplementary Note 3). An additional constraint on maximum thickness is provided by the lower temperature limit of 120 °C for biomarker reaction: thicker faults cannot attain peak temperatures above this limit for an earthquake like Tohoku-Oki because of their larger volume (Supplementary Fig. 7). Minimum fault thickness is impossible to constrain from core images due to their resolution. We instead constrain the minimum thickness of all modeled faults using a maximum temperature limit based upon the absence (to date) of reported pseudotachylyte or clay amorphization throughout the JFAST core[18,41]. Using this constraint, fault thicknesses that yield temperatures above 900 °C are excluded (Supplementary Fig. 7)[42,43]. Any amorphous material in the JFAST core would likely be present only on very thin features, and it is possible that finer-scale sampling in the future will reveal amorphous material. However, even if temperatures higher than 900 °C are possible, many large earthquakes would still be required to explain the biomarker signals. This would only change the minimum thickness and slightly lower the minimum number of possible earthquakes or minimum slip distance each fault has experienced (see Supplementary Note 3).

In addition to fault thickness, we use core observations to determine the distance between each sample and their adjacent faults (Supplementary Fig. 6 and Supplementary Data 3). Fault thickness and the distance between a sample and a seismic fault interface provide strong constraints on whether the model can reproduce the observed biomarker anomalies (Fig. 3c). As the fault thickness increases, or the distance from the fault increases, more slip is required to explain the biomarker reactions. Therefore, samples from thicker faults or samples located further

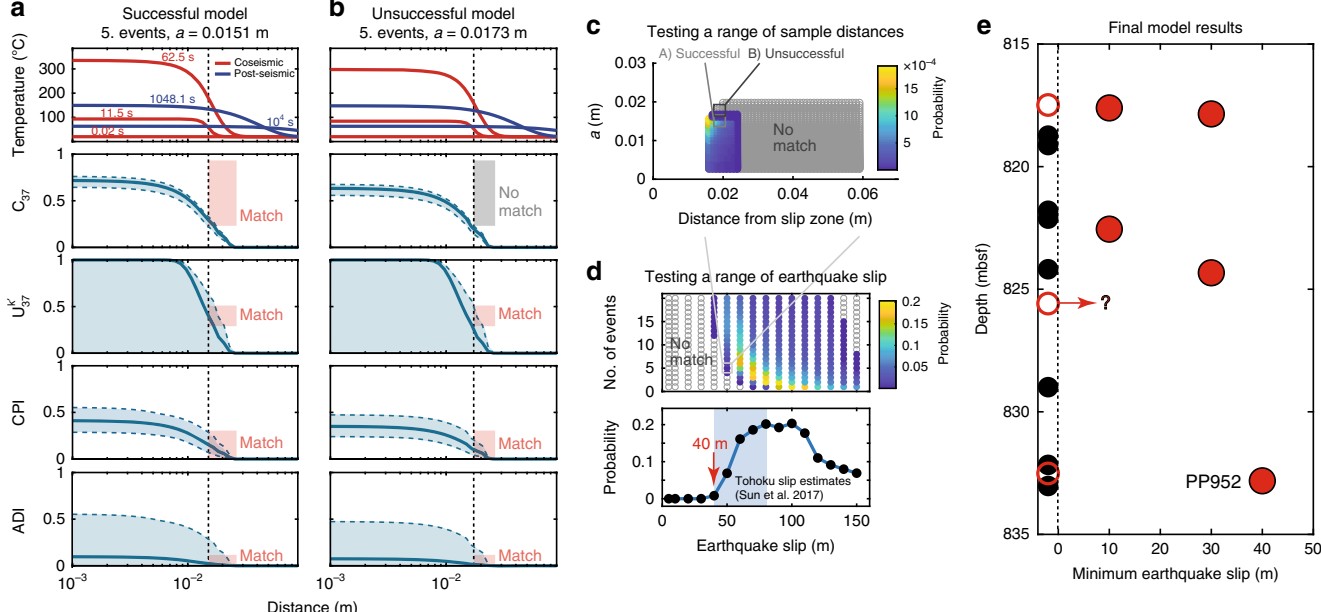

**Fig. 3 Modeling slip to match biomarker measurements.** The forward modeling procedure considers variation in fault half width, slip magnitude, number of earthquakes, and distance of a sample from the fault. Here, we show an example for sample PP952, assuming five earthquakes with slip of 50 m. **a** A successful model fit for all biomarkers using $a = 0.0151$ m. The first panel shows the temperature with distance from the fault for different time-steps. Lower panels show the final biomarker fraction reacted with distance from the fault (heavy lines) with blue shaded regions showing uncertainty in biomarker kinetics. Measured range of each biomarker anomaly is shown as boxes (height). The width of the boxes represents the sample width, limited by the constraint that the sample is outside of the slipping zone. Boxes are shaded red for successful matches and gray if unsuccessful. **b** Model is unsuccessful for $a = 0.0173$ m due to a mismatch between modeled and measured $C_{37}$ total fraction reacted. **c** Plot of distance from the fault vs. half width with hotter colors showing higher probability of a successful model (successful models/total model runs). Gray circles indicate unsuccessful models ($P = 0$). **d** Lower panel shows the probability of a successful model at the range of earthquake slips tested. Minimum slip with a nonzero probability is indicated with the red arrow. Range of slip estimates for the Tohoku-Oki earthquake shown by the blue shaded region[4]. Upper panel shows the slip vs. number of earthquakes with probability indicated by color as in **c**. **e** For samples with biomarker anomalies we find the minimum slip distance of a repeating event (up to 3.2 km of total displacement) needed to match the biomarker anomalies. These modeling results indicate that any of the faults with heating anomalies could have hosted the 2011 Tohoku-Oki earthquake, and that all observed biomarker anomalies require earthquakes with a minimum of 10 m slip per event. Red symbols indicate features with clear biomarker anomalies. Hollow red symbols represent sample PP948, which has alkenone concentrations below the quantification limit and is not modeled (Supplementary Note 3) and samples PP944 and PP951 for which structural evidence is ambiguous (Supplementary Note 3).

from a fault may require larger or more numerous earthquakes than samples with greater biomarker reaction that are located within a thin fault (Figs. 2 and 3d).

To determine which faults could have hosted the Tohoku-Oki earthquake, we forward model an event with 50 m slip[4,12,14,15] using the shear stress (0.54 MPa) inferred from the JFAST temperature observatory[23], and a slip velocity of 1 m/s inferred from geodetic observations of the shallow slip at Tohoku[40] ("Methods", Supplementary Fig. 8). We find that any of the faults with heating anomalies at JFAST could have hosted at least one Tohoku-Oki-sized earthquake (Supplementary Figs. 9–10). For many of these faults, additional earthquakes with large slip are needed to generate the observed biomarker signal (faults at 817.85, 822.55, and 832.815 mbsf).

Although we find that every fault with a thermal maturity anomaly could have hosted the Tohoku-Oki earthquake, this is not a unique solution: multiple smaller earthquakes might also generate the cumulative biomarker maturity anomalies that we observe (Supplementary Figs. 10 and 11). To understand the absolute minimum earthquake size that could produce the observed anomalies, we model multiple earthquakes of equal slip up to the number of events that sum to 3.2 km of total displacement on the plate boundary as this represents an upper bound on the number of ruptures of a certain size that could have occurred[13]. If the observed anomalies cannot be reproduced by

the cumulative effect of these earthquakes, the fault represented by that sample must have experienced at least one earthquake of larger magnitude (Supplementary Data 3 and Supplementary Fig. 10). We model increasingly large earthquakes until we find an earthquake slip capable of reproducing the observed biomarker anomalies (Fig. 3e). This does not mean that earthquakes larger than this size did not occur on these faults, just that they are not required to explain the thermal maturity anomalies. The minimum event slip is useful because it means that earthquakes at or below this size do not significantly contribute to the cumulative biomarker thermal maturity of the fault, due to their lower temperature rise and the non-linear nature of the reaction kinetics. We find that for all modeled samples, the biomarker signal is generated by earthquakes with more than 10 m of slip (~$M_w$8 and larger[44]), and on some faults earthquake slips up to 40 m are required (Fig. 3e).

These results indicate that all of our modeled biomarker maturity anomalies are caused by large events rather than the accumulation of many small events. Reasonably assuming similar shear stress to the Tohoku-Oki event, these results imply that large earthquakes are commonplace in the shallow regions of the Japan Trench. If the shear stress during slip for past earthquakes differed from the estimates for the Tohoku-Oki earthquake, our model results would change as well. However, experiments show that the steady-state friction at high velocity is generally

extremely low in wet clay-rich gouges such as found at JFAST[9,21,45,46]. Such low friction values would yield low shear stresses, comparable to the value of 0.54 MPa[23] that we apply, and would lead to similar modeling results.

## Discussion

Our analysis allows us to assess which (if any) fault parameters control where shallow slip occurs at the Japan Trench. Slip might be expected to localize within the weakest lithologies[9,13]; however, seismic faults are present in almost all of the major lithologies recovered in the JFAST core despite large variations in friction values across JFAST lithologies[7]. A low peak in strength at the onset of fast slip has also been hypothesized as a way to propagate shallow slip[9,21,46]. Only two lithologies at the Japan Trench have been tested at high velocities where this friction peak can be measured[9,11,21]. These experiments show that the pelagic clay (Unit C2) has a smaller peak in strength compared with the mudstone (Unit A1). However, our results show seismic slip in both lithologies (Fig. 4), suggesting that differences in peak strength may not control shallow seismic rupture propagation at the Japan Trench.

Theory predicts that slip will localize at boundaries in compliance[48]. At the Japan Trench, localized earthquake slip is found at multiple structures within a 15-m-thick faulted layer which itself is located above the boundary between more clay-rich

sediments and the underlying cherts (and possibly basalts). This boundary stands out as the largest discontinuity in material properties, and may be the overarching control on where the plate interface occurs. Geometric, material property, and interaction effects may then determine which individual fault(s) slip in this volume during an earthquake. In this case, the strength of each fault should not be considered individually. The energy feeding a propagating rupture comes from the crustal volume around the fault that has been strained, which is much wider than the fault zone[49], and multiple fault strands could slip in a single earthquake[50]. The interaction between strands could enhance or retard shallow slip on any particular strand[51]. This could mean that energy is minimized on a system level rather than an individual fault level[52].

The convergence rate at the Japan Trench is ~9 cm/year[53], and the JFAST core was drilled through rocks that have experienced ~3.2 km of convergence[13], suggesting ~36,000 years of deformation. Our biomarker analysis uniquely illustrates that the propagation of great earthquakes into the shallow reaches of the subduction zone occurred repeatedly, building on the evidence of three paleotsunamis in the area[5] and suggesting a long-lived seismic hazard. Determining thermal histories of fault rocks in other subduction zones will require new fault zone drilling to provide similar paleoseismic records[25], which are useful for understanding shallow tsunamigenic seismic slip.

## Methods

**Samples**. We sampled the JFAST core throughout its recovered depth (183–833 mbsf, sample spacing ranging from 0.7–507 m) with finer sample spacing near the bottom of the core (817–833 mbsf, sample spacing ranging from 0.12–1.6 m), where multiple faults are present[13,17–19]. DSDP Site 436 was selected as a reference for incoming sediments at JFAST due to its proximity to the JFAST site[47]. One of the most important components of the biomarker analysis is determining the initial biomarker content of the faulted sediments. Analysis of trace elements shows that western Pacific sedimentary units are broadly consistent over large distances[17]. Therefore, by correlating the chemostratigraphy between the JFAST and 436 sites, we calibrate the range of initial organic content for each sedimentary unit.

**Quantification of biomarker concentrations**. Biomarker concentrations were determined following methods described in Rabinowitz et al.[33] Sediment was freeze dried at a vacuum of 6 Pa (60 μbar) and then crushed in a mortar and pestle that was solvent-rinsed with dichloromethane (DCM) and methanol (MeOH). The total lipid extract (TLE) was obtained through sonication extraction using a solution of 9:1 DCM:MeOH with three 15-min sonications. In order to ensure that all extractable organic material was analyzed, a second extraction of the sediment was performed using an accelerated solvent extraction system (ASE) following the methods of Rabinowitz et al.[33], again with an extraction solvent of 9:1 DCM: MeOH. The ASE extractions were conducted at 100 °C, which has been demonstrated to effectively extract organic material without degrading the biomarkers analyzed in this study[33].

Once the sediment was extracted, 50 μl of a recovery standard containing 5α-androstane and stearyl stearate were added to each TLE. The TLEs were then evaporated under $N_2$, transferred into a 4 ml vial using DCM, and dried down again. TLEs were separated into three fractions (aliphatic, ketone, and polar) using silica gel column chromatography. The F1 (aliphatic) fraction was obtained by pipetting the sample in 1 ml of hexane into a Pasteur pipette column half filled with DCM-rinsed silica gel that had been activated at 100 °C for >24 h. An additional ~3 ml of hexane was pipetted onto the column to elute the aliphatic fraction. This procedure was repeated using DCM and MeOH to separate the F2 (ketone) and F3 (polar) fractions, respectively. The F1 and F2 fractions were evaporated and transferred to 2 ml vials using DCM. These were then evaporated and brought up in hexane (F1) and toluene (F2) for analysis by gas chromatograph. At the Lamont–Doherty Earth Observatory, n-alkanes (F1) were analyzed using an Agilent gas chromatograph with a mass selective detector (GC–MSD) and alkenones (F2) were analyzed with a thermo gas chromatograph with a flame ionization detector (GC–FID). TLEs from sonication and ASE extractions were analyzed separately (see below). Total n-alkane concentrations in a sample were obtained by summing the concentrations of each molecule determined in the sonication and ASE extracts. Alkenone concentrations in the ASE extracted fraction of the samples were found to be below the detection limit and only sonication extractions were used for analysis.

The GC–MSD was run with a multimode inlet using a 30 m DB-5 column (0.25 mm i.d., 0.25 μm phase thickness). One microliter of sample was injected and the oven was held at 60 °C for 1.5 min. The temperature was ramped to 150 °C at

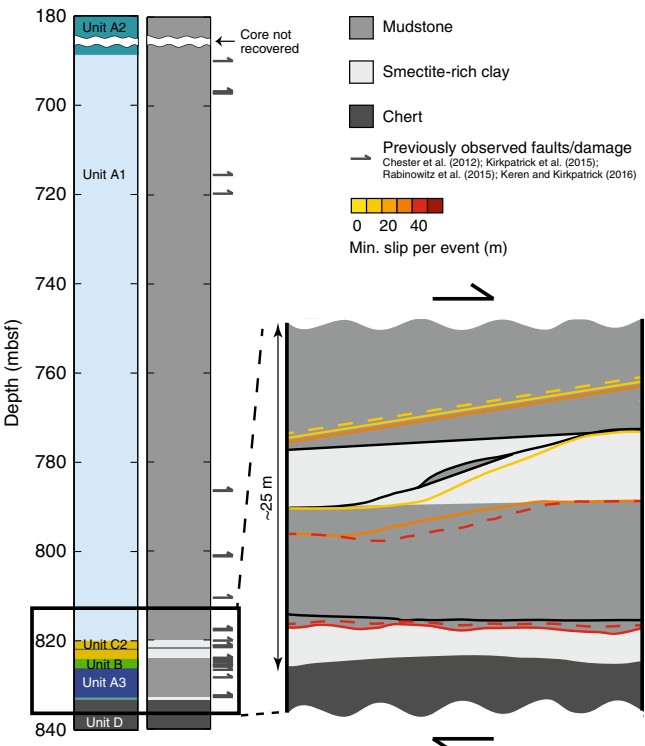

**Fig. 4 Schematic of the structure recovered at JFAST[17,18].** Trace element stratigraphy of the JFAST core is shown on the left with sedimentary units colored and labeled as in Fig. 1[17]. Generalized stratigraphy is shown in the middle with mudstone, smectite-rich clay, and chert lithologies indicated in different shades of gray. Previously observed faults and damage[17–19, 47] are indicated on the right side of the stratigraphic column. On the right, a zoomed-in schematic of the structure recovered at JFAST[17,18] is shown and faults with biomarker thermal anomalies are colored by the minimum slip magnitude capable of reproducing the observations within the allowable amount of total slip[13]. Dashed lines represent PP948, which was not modeled as well as samples PP944 and PP951 for which structural evidence is ambiguous.

15 °C min$^{-1}$ and then to 320 °C at 4 °C min$^{-1}$ followed by a 10-min hold. The injector was held at 60 °C for 0.1 min and then ballistically ramped to 320 °C where it was held for the duration of the GC run. Chromatograms were quantified using Chemstation software. The Thermo Trace GC Ultra GC–FID was run using a PTV injector with a 2 mm i.d. silicosteel liner and a 60 m × 0.250 mm i.d. DB-1 column with a stationary phase thickness of 0.1 μm and a 10 m × 0.250 mm nonpolar deactivated retention gap. One microliter of sample in toluene was injected, after which, the oven was held at 90 °C for 1.5 min, raised to 250 °C at 25 °C min$^{-1}$, then raised to 313 °C at 1 °C min$^{-1}$ and finally raised to 320 °C at 10 °C min$^{-1}$ and held for 20 min. To quantify alkenone concentrations, we integrated the chromatograms from the F2 fraction using ChromQuest software.

We analyzed an $n$-alkane drift, a mixture of C$_8$–C$_{40}$ $n$-alkanes containing the 5α-androstane standard, along with the F1 fractions of the samples. This $n$-alkane drift is used to calculate a response factor for each $n$-alkane homolog relative to 5α-androstane. We multiply the area ratio of each $n$-alkane molecule to 5α-androstane by that molecule's response factor to obtain the amount ratio of each $n$-alkane molecule relative to the 5α-androstane standard added to that sample. F2 fractions of samples were analyzed with a stearyl stearate standard contained within the recovery standard added to each sample. The amount ratio of alkenone molecules is determined by dividing the area of each alkenone molecule peak by the area of the stearyl stearate standard. Concentrations are calculated by multiplying the amount ratio of each molecule by the amount of the respective standard added to the sample and dividing by the sample weight (Supplementary Data 1).

We report C$_{12}$–C$_{35}$ $n$-alkane concentrations as these were the molecules with concentrations high enough to reliably quantify (Supplementary Data 1). We use these concentrations to calculate the $n$-alkane parameters used in the analysis (Supplementary Data 2). The CPI is calculated by dividing the summed concentrations of odd chain-length $n$-alkanes by the summed concentrations of even chain-length $n$-alkanes between chain lengths of 26–35 ($\Sigma$C$_{odd,27-35}$/$\Sigma$C$_{even,26-34}$). The ADI, defined by Rabinowitz et al.[33], is calculated as (C$_{27}$ + C$_{31}$)/(C$_{28}$ + C$_{29}$ + C$_{30}$). CPI is repeatable to <1.5% and ADI is repeatable to <1% (1 s)[33]. We calculate the alkenone concentration by adding the concentrations of the measured C$_{37}$ alkenone molecules (MK37:3 and MK37:2). $U^{k'}_{37}$ values are determined as (MK37:2)/(MK37:2 + MK37:3) (Supplementary Data 2). Alkenone concentrations are repeatable to 4.1% and $U^{k'}_{37}$ values are repeatable to 0.0033 (1 s)[33].

**Total organic carbon**. After the TLE was extracted from each sample, about 1 g of sediment was set aside to measure total organic carbon (TOC). The dried sediment was transferred to a weighed 50 ml centrifuge tube and the tube was weighed again to determine the sample size. About 20 ml of 1 N HCl (1:11 12 N HCl:ultra-pure distilled water) was added to each tube to dissolve any carbonate. The tubes were shaken by hand and then using a vortex mixer and allowed to sit for 2 h. Another 20 ml of HCl was then added and the tubes were shaken again and allowed to sit overnight.

Tubes were then filled the rest of the way with ultra-pure distilled water and centrifuged for 15 min. The supernatant was carefully decanted and the rinsing procedure was repeated until the pH of the liquid after centrifugation was about equal to the rinse water, typically six rinses (after the third rinse, distilled water from the tap was used). The pellets at the bottom of the vials were then freeze dried for one day. Tubes were again weighed to determine the amount of sample lost during the decarbonation procedure. Small amounts of sample (10–50 mg) were weighed into aluminum boats and TOC was measured on a Costech Elemental Analyzer. Because the samples have low organic carbon concentrations, TOC was measured at the H8 sensitivity setting on the EA's thermal conductivity detector (Supplementary Data 1). Relative uncertainty in TOC values is ~4% of the reported value based upon replicate analysis of 15 samples.

**Calculation of fraction reacted**. To determine whether thermal alteration has occurred, we compare JFAST biomarker concentrations to the biomarker concentrations of undeformed sediments in samples from corresponding sedimentary units in the reference core. Samples were correlated to sedimentary units at Site 436 using the trace element stratigraphy developed by Rabinowitz et al.[17]. To calculate the biomarker reaction, the fraction remaining ($r$) was determined by dividing the alkenone concentration, $U^{k'}_{37}$ value, CPI or ADI of the JFAST sample by corresponding biomarker parameter values in the protolith unit measured at Site 436. For each JFAST sample, $r$ values were calculated with respect to each sample in the correlated Site 436 unit. The range of these fraction remaining values are plotted in Fig. 2 as box plots with the median value of the fraction remaining indicated by the vertical line, the boxes corresponding to the quartiles (Q$_1$ = 25th and Q$_3$ = 75th percentiles) and the whiskers to values lying ~2.7σ from the median. Outliers, plotted as individual data points in Fig. 2 are values that are less than Q$_1$ − 1.5*(Q$_3$ − Q$_1$) or greater than Q$_3$ + 1.5*(Q$_3$ − Q$_1$).

**Modeling of temperature rise on faults**. An integrated time-temperature history of a shear-heating event is used to analyze biomarker alteration in natural fault samples[38]. In order to account for the duration of elevated temperatures with distance from the observed faults, we use a fault-heating model that includes heat diffusion away from the fault. In this model slip velocity, fault half-width, shear

stress, and sedimentary material properties are prescribed and held constant:[22]

$$\theta(x < a, t) = \frac{\tau}{\rho c} \frac{v}{2a} \left\{ \begin{array}{c} t \left[ 1 - 2i^2 erfc \frac{a-x}{\sqrt{4\alpha t}} - 2i^2 erfc \frac{a+x}{\sqrt{4\alpha t}} \right] \\ -H(t - t^*)(t - t^*) \left[ 1 - 2i^2 erfc \frac{a-x}{\sqrt{4\alpha(t-t^*)}} - 2i^2 erfc \frac{a+x}{\sqrt{4\alpha(t-t^*)}} \right] \end{array} \right\},$$

$$\theta(x > a, t) = \frac{\tau}{\rho c} \frac{v}{2a} \left\{ \begin{array}{c} t \left[ 2i^2 erfc \frac{x-a}{\sqrt{4\alpha t}} - 2i^2 erfc \frac{x+a}{\sqrt{4\alpha t}} \right] \\ -H(t - t^*)(t - t^*) \left[ 2i^2 erfc \frac{x-a}{\sqrt{4\alpha(t-t^*)}} - 2i^2 erfc \frac{x+a}{\sqrt{4\alpha(t-t^*)}} \right] \end{array} \right\}$$

(1)

where θ is the temperature rise (K), a is the fault half width (m), τ is shear stress on the fault (Pa), t is time (s), $t^*$ is the slip duration (s), x is the distance from the center of the fault (m), ρ is the density of the material (kg m$^{-3}$), c is the heat capacity (J kg$^{-1}$ K$^{-1}$), v is the slip velocity (m s$^{-1}$), α is the thermal diffusivity (m$^2$ s$^{-1}$), $i^2erfc$ is the second integral of the complementary error function, and H (ς) is the Heaviside function which is evaluated for ς = t − $t^*$ so that the multiplied terms on the right are only applied when t ≥ $t^*$[54]. Absolute temperature is determined by adding θ to the background temperature, which is 20 °C (293.15 K) at ~700 mbsf at the JFAST site[23]. Models were run to 10,000 s, whereas the longest earthquake duration modeled in this study is 150 s for earthquakes with 150 m slip.

We note that this model does not take into account possible advection of fluids, which have been inferred to play an important role in heat and stress transfer during earthquakes in some fault zones, indicated by the presence of features such as mineral veining around faults[55–58]. However, such features are not observed in the JFAST core[18,19]. Indeed, the presence of a temperature signal more than one year after the earthquake implies that advection was limited[23]. In addition, faults directly around 820 mbsf do not experience transient fluid advection in response to regional earthquakes in contrast to transient advective signals (<~0.1 °C) observed at shallower depths[59]. Here, we model the temperature rise generated only through seismic slip and heat diffusion on observed structures in the JFAST core. Advection of hydrothermal fluids immediately after seismic slip would serve to transport heat to further distances from the slipping surface. For samples within the slipping zone, this effect would lower the measurable peak temperature at the slip surface and Eq. 1 would underestimate the temperature rise. Conversely, for samples outside of the slipping zone, hydrothermal fluid advection would yield a higher temperature further from the fault compared with values attained through diffusion alone. In these cases, Eq. 1 would overestimate the peak temperature at the center of the slipping zone.

Biomarker thermal alteration is modeled using an expanded form of the Arrhenius equation:

$$f = 1 - \exp\left\{ -At \times \exp\left\{ -\frac{E_a}{RT} \right\} \right\},$$

(2)

where $f$ is the fraction reacted (1 – $r$, where $r$ is the fraction remaining), $A$ is the pre-exponential frequency factor (s$^{-1}$), $E_a$ is the activation energy (J mol$^{-1}$), $R$ is the gas constant (8.314 J K$^{-1}$ mol$^{-1}$), and $T$ is temperature (K). Note that for $U^{k'}_{37}$, which increases with increasing thermal maturity, f is calculated as:

$$f_{U^{k'}_{37}} = 1 - \frac{1 - U^{k'}_{37,JFAST}}{1 - U^{k'}_{37,Site436}}.$$

(3)

The time-temperature history for earthquakes with slip magnitudes ranging from 5–150 m in 10-m increments is calculated using Eq. 1, for a range of possible fault widths consistent with structural observations for each sample. These time-temperature histories are combined with the experimentally determined kinetic parameters of biomarker thermal maturation[33] to determine the predicted fraction of the biomarkers reacted for each half width. In addition, model f values are calculated for scenarios incorporating multiple earthquakes. For each half width, we consider 100 random samples from the joint probability distribution of $E_a$/A pairs that define the uncertainty envelope on the biomarker kinetic parameters determined by Rabinowitz et al.[33] and consider the degree of biomarker alteration resulting from repeated earthquakes with cumulative displacement limited to 3.2 km[13]. Calculated f values are compared with the measured f values for all biomarker parameters in each sample (e.g. Fig. 2). For each biomarker parameter in a sample there is a range of measured f values due to the variation in the initial, unaltered values within each sedimentary unit at Site 436. A comparison between the model and the data are considered successful if the calculated f values within a sample (distance from fault ± the half of the width of the sample) fall between the minimum and maximum measured f values for all biomarker constraints in that sample. We note that, due to the large uncertainty in the $U^{k'}_{37}$ kinetics, $U^{k'}_{37}$ provides the least constraint of the four biomarker parameters investigated here.

**Treatment of model results**. Our goal with the modeling is to use the measured biomarker parameters to constrain the possible earthquake magnitudes and number of earthquakes that a sample could have experienced. For a particular earthquake magnitude (e.g., 50 m for a Tohoku-sized event), we model how combinations of fault half widths ($a_i$, $i = 1…m$) and numbers of earthquakes ($N_j$, $j = 1…n$) affect biomarker parameters ($H$). For each $a_i$, $N_j$ pair we run models

$k = 1\ldots100$ with random samples of the joint probability distribution of activation energy and frequency factor $(E, A)_k$ for each biomarker parameter $(H(l), l = U_k, C_{37}tot, CPI, ADI)$. For each biomarker parameter, $H(l)$, in each model, $k$, we ask whether the modeled biomarker parameter $(H_{model})$ matches the measured biomarker value of the sample $(H_{meas})$. Each measured biomarker parameter $H_{meas}(l)$ has a range $H_{min}(l)$ to $H_{max}(l)$ of possible values reflecting its uncertainty due to the uncertainty in the initial, unaltered value. Each modeled biomarker parameter $H(l)$ varies with distance $x$ from the fault, $H_{model}(x, l)$, and therefore also has a range of values across the physical width $x_{min}$ to $x_{max}$ of the sample (the distance from the fault to the closest and furthest edges of the sample) with a given number of distances analyzed in that range ($W_{sample}$). A match is successful if the range of measured biomarker values and the range of modeled biomarker values in a sample overlap (Fig. 3 and Supplementary Fig. 8):

$$\text{match}(k, l) = \begin{cases} 0, & \{\emptyset\} = \{H_{meas}|H_{min} \le H \le H_{max}\} \bigcup \{H_{model}(x) : x_{min} \le x \le x_{max}\} \\ 1, & \text{otherwise} \end{cases}$$

(4)

From the number of successful matches, we calculate a probability that a model matches the data for each biomarker parameter $l$:

$$p(l) = \frac{\sum_{k=1}^{100} \text{match}(k, l)}{100 \times W_{sample}}.$$

(5)

The overall probability that a model matches the data ($p_{all}$) is given by the product of the individual probabilities for each biomarker parameter:

$$p_{all} = p_{U_{37}^{k'}} p_{C_{37}tot} p_{CPI} p_{ADI}$$

(6)

In many cases, there is a range of distances from a fault that a sample could occupy due to uncertainty in the exact location of the slip zone, and we calculate $p_{all}$ as a function of the sample distance from the fault. For each distance, we consider the sample width as above by adjusting $x_{min}$ and $x_{max}$ together to maintain a constant width, and then calculate match($k, l$), $p(l)$, and $p_{all}$ (Eqs. 4–6, Fig. 3c). We take the maximum value of $p_{all}$ across all distances as the probability of a match for that particular half width ($a_i$), and number of earthquakes ($N_j$) pair (Fig. 3d). Supplementary Fig. 9 shows the match probability for a 50-m slip event as a function of the fault half width, and number of earthquakes. It is apparent that all of the samples with a biomarker heating anomaly could be matched by one or more 50-m events.

We investigate how different slip distances affect the modeled biomarker maturity signal by repeating the above steps for different displacements. The probability of a given slip distance is then taken to be the maximum of the calculated probabilities across all $m \times n$ pairs of half width and numbers of earthquakes for that magnitude. Supplementary Fig. 10 shows these probabilities as a function of slip distance for each biomarker sample. It is apparent from these plots that for each sample there is an event slip below which no models can match the biomarker data ($p_{all} = 0$). The next higher tested slip distance is then taken as the minimum slip distance required to match the biomarker parameters.

## Data availability

All data are available in the supplementary tables of this paper.

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

## Acknowledgements

Funding was provided by NSF OCE 12-60555 to H.M.S., P.J.P., OCE 12-60602 to J.D.K., and a Schlanger Ocean Drilling Fellowship and NSF GRFP (DGE-11-44155) to H.S.R. Instrumentation for biomarker analyses was supported by the Columbia Center for Climate and Life. This research used samples and data provided by the Integrated Ocean Drilling Program (IODP). Images and core description logs used in this study are available for download from IODP. The authors thank IODP and the Kochi Core Center for help obtaining samples, R. Sheppard for help with laboratory work, and M. Pratt for help with modeling. Early versions of the paper were improved by comments from Patrick Fulton.

## Author contributions

H.S.R. conducted laboratory work to measure biomarkers in core samples and modeling and H.M.S. and P.J.P. initially conceived the project. H.S.R., H.M.S., P.J.P., C.D.R. and J.D.K. contributed to data analysis and the writing of this paper.

## Competing interests

The authors declare no competing interests.
