## [Peer Review File · Nature Communications]

Reviewers' comments:

Reviewer #1 (Remarks to the Author):

This manuscript presents a results of biomarker analysis of core samples collected around a fault zone of the 2011 Tohoku-Oki earthquake, and find a clear evidence indicating that multiple faults in different lithology have hosted multiple large earthquakes. This manuscript concludes that lithologic contrasts in friction do not necessarily determine the likelihood of large shallow slip. This conclusion brings a new concept to understand a fault process of an in a shallow part of a subduction zone. Because, following results of a high velocity frictional experiment using the JFAST core, a weak-clay layer has been considered to be a unique lithological factor to host a large slip at the shallow part of the subduction zone.

I cannot assess an accuracy of the results of this study, since I am not an expert of the method this study used, I believe that other reviewers evaluate the method and accuracy of the results. Assuming the results are theoretically and methodologically correct, the results indicating multiple faulting in the different lithological units brings a crucial new idea to examine a shallow fault slip in subduction zone and consequently a generation of large tsunami.

I am convinced, based on the results, that there are several earthquake faults within the hemipelagic mud layer in addition to the pelagic clay layer. However, stating that "lithologic contrasts in friction do not necessarily determine the likelihood of large shallow slip" might be overstated. In a large-scale structural point of view, there are many branching (or secondary) faults in the frontal wedge branching from a main fault immediately above the oceanic crust, as one can see in Fig. 1B of this manuscript. If this faulting system (i.e., branching fault from the primally fault above the oceanic crust) can be applicable in the core-sample scale, it could be possible to interpret that faults in the hemipelagic mud layer can be secondary faults (or branching faults) branching off from a primally fault in the weak-clay layer. If this idea is acceptable, a weak-clay layer could be a lithological factor to host a large slip in the shallow part of the subducton zone. The authors may discuss this point to make their conclusion more confident.

Reviewer #2 (Remarks to the Author):

Paper rationale:

The authors look for thermal signatures of past earthquake ruptures on an active thrust fault. Relatively high temperature transients can be achieved on faults due to frictional heating, but only if the slip is relatively fast (typically > 0.1 m/s). Fast slip is a condition for seismic slip, therefore traces of transient temperature surges can generally be interpreted as a sign of seismic activity on a given fault. Recognising the potential for large seismic slip on an active fault has clear implications for risk assessment and hazard mitigation.

An accepted seismic thermal signature is solidified frictional melt (pseudotachylyte), but it is not pervasive, especially on the superficial and accessible segments of currently active faults.

Therefore identifying and using signatures other than melt has a potential impact on seismology and risk assessment.

In this case, the authors illustrate a novel use of chemical de-bonding and carbon reactions triggered by thermal heating within organic matter which can be found on the topmost portion of subduction zone large faults. They apply this method to a fault which is active and recently caused major hazard in Japan. The specific aim here is to infer the number of large earthquakes that may have occurred in the past centuries, producing results that can feed into the risk mitigation for the concerned geographical area(s), in particular regarding superficial slip and the tsunamigenic potential of the fault system.

The authors model the possible thermal surge amplitude and duration on the studied fault zone as a function of the seismic slip amount and of the chemical degradation kinetics, on several fault strands in the accretionary prism, seeking which ones show evidence of large seismic slip as a function of the inferred thermal surge.

Their interpretation is that at least 8 fault strands have indeed undergone large (in excess of 10 m) superficial seismic slip.

Methods and Rigour:

The detail of the chemical kinetics described here is beyond the reach of my knowledge, but it appears that the methodology is sound and well justified in thorough methods section and sup. Mat.

Attention is paid to the crucial error margin of the measurements and the uncertainty of the model, in a way that suggests rigour. Importantly, the measurements performed on core samples for the IODP borehole are not absolute measurements of the decomposition reaction, but relative to an undisturbed section of the same unit off-fault (other borehole).

This allows to mitigate the risk of misinterpreting chemical fluctuations within the geological sequence due to other causes than the co-seismic heating.

In regards to the seismic slip mechanicals and thermal diffusion model used to calculate the heating as a function of slip, the approach is fine, although I have a few minor clarifications to ask (see below).

Recommendation:

I would recommend to publish this paper, after dealing with a few minor remarks below.

Questions, edits and possible clarifications:

The authors use the ratio r of chemical fraction between JFAST samples and Site436 borehole, which is shown in figure 2. I would also show a reference of variability (in the form of an error bar or std deviation) between the different analysed samples within the Site436 alone. This would allow to appraise the signal to noise ratio in r , and how much it differs from the random fluctuations within the geological section, with no further work and only a minor addition in the manuscript (possibly on Figure 2).

Line 149. Eight faults show signatures compatible with large seismic slip, but how many faults have been analysed in total? It is not easy to infer the exact total number from the figures (although it may be indicated elsewhere in the text).

I could not find a mention of the possible time span of seismicity which is covered by this investigation. Is it possible to give a upper and lower bound ?

Is each of the 8 “large slip” faults interpreted as an individual earthquake, or possibly there were fewer than 8 earthquakes but the rupture splayed on on multiple faults, or is it not possible to discern?

Equation S1 re the thermal model. There is no factor of 2 in the original equation of cited work. Is it a matter of half-thickness versus whole thickness, or just a typo? Also the equation represents the evolution of temperature after slip cessation. Therefore it neglects the temperature rise phase integrated over the seismic slip duration in the chemical kinetics. This may be negligible, if the duration of the heated state (diffusion time) is large with respect to the slip duration. However, this probably should be indicated in the corresponding paragraph.

Minor edits:

The source in Table S3 for the co-seismic shear stress is not reference 16(Chester) but reference 23(Fulton).

Figure 1 legend: replace “...regions of equal slip...” with “regions of equal slip as inferred from inversion of geophysical data...”

Reference 22: add “USGS open file report 86508”

With Best Regards,
Stefan Nielsen

Reviewer #3 (Remarks to the Author):

This is a most novel application of molecular thermal maturation indicators. In this study, a core (JFAST) taken in a region with multiple fault slips is compared to a nearby DSDP core where the faults are absent. The stratigraphy of the cores are closely correlated so that baseline values for the thermal parameters can be obtained. The JFAST core has evidence of multiple seismic slippages as evidenced by localized frictional heating. The degree of unsaturation of alkenones and the carbon preference of plant derived n-alkanes are examined and the kinetic models, determined in prior studies, for these reactions are used to constrain the heating events during earthquake slippage. This approach, to my knowledge, is unique and the influence of localized seismic heating on biomarkers has never been described. In general, the authors have tried to address the basic aspects of the problem and have tried to minimize likely error. The most obvious is establishing the starting

(non-seismic) value from which the extent of reaction(s) are determined. I appreciate the care in which the authors have correlated the sediments between the DSDP and JFAST cores to determine these initial values. However, the CPI of immature sediments is so closely dependent on the influence of organic matter input that I'm not sure that the DSDP cores, regardless of elemental matches, accurately reflect the starting values. I suggest the authors add a paragraph or two that would explain the influence in the uncertainty of the initial values on the overall results. Values reported in Table S1 are being reported at a higher level of significance than is justified.

All in all, an excellent study showing an new application of molecular markers, one certainly worthy of publication

Clifford Walters

We would like to thank the reviewers for their thoughtful comments. Our replies are shown in blue.

Reviewers' comments:

Reviewer #1 (Remarks to the Author):

This manuscript presents a results of biomarker analysis of core samples collected around a fault zone of the 2011 Tohoku-Oki earthquake, and find a clear evidence indicating that multiple faults in different lithology have hosted multiple large earthquakes. This manuscript concludes that lithologic contrasts in friction do not necessarily determine the likelihood of large shallow slip. This conclusion brings a new concept to understand a fault process of an in a shallow part of a subduction zone. Because, following results of a high velocity frictional experiment using the JFAST core, a weak-clay layer has been considered to be a unique lithological factor to host a large slip at the shallow part of the subduction zone.

I cannot assess an accuracy of the results of this study, since I am not an expert of the method this study used, I believe that other reviewers evaluate the method and accuracy of the results. Assuming the results are theoretically and methodologically correct, the results indicating multiple faulting in the different lithological units brings a crucial new idea to examine a shallow fault slip in subduction zone and consequently a generation of large tsunami.

I am convinced, based on the results, that there are several earthquake faults within the hemipelagic mud layer in addition to the pelagic clay layer. However, stating that “lithologic contrasts in friction do not necessarily determine the likelihood of large shallow slip” might be overstated. In a large-scale structural point of view, there are many branching (or secondary) faults in the frontal wedge branching from a main fault immediately above the oceanic crust, as one can see in Fig. 1B of this manuscript. If this faulting system (i.e., branching fault from the primally fault above the oceanic crust) can be applicable in the core-sample scale, it could be possible to interpret that faults in the hemipelagic mud layer can be secondary faults (or branching faults) branching off from a primally fault in the weak-clay layer. If this idea is acceptable, a weak-clay layer could be a lithological factor to host a large slip in the shallow part of the subducton zone. The authors may discuss this point to make their conclusion more confident.

We fully agree with the reviewer that the plate boundary is primarily located at the interface between the sediments and the chert/basalt because of the high compliance contrast between these rock types. Lines 311 to 317 make this point.

Reviewer #2 (Remarks to the Author):

Paper rationale:

The authors look for thermal signatures of past earthquake ruptures on an active thrust fault. Relatively high temperature transients can be achieved on faults due to frictional heating, but only if the slip is relatively fast (typically > 0.1 m/s). Fast slip is a condition for seismic slip, therefore traces of transient temperature surges can generally be interpreted as a sign of seismic activity on a given fault. Recognising the potential for large seismic slip on an active fault has clear implications for risk assessment and hazard mitigation.

An accepted seismic thermal signature is solidified frictional melt (pseudotachylyte), but it is not pervasive, especially on the superficial and accessible segments of currently active faults.

Therefore identifying and using signatures other than melt has a potential impact on seismology and risk assessment.

In this case, the authors illustrate a novel use of chemical de-bonding and carbon reactions triggered by thermal heating within organic matter which can be found on the topmost portion of subduction zone large faults. They apply this method to a fault which is active and recently caused major hazard in Japan. The specific aim here is to infer the number of large earthquakes that may have occurred in the past centuries, producing results that can feed into the risk mitigation for the concerned geographical area(s), in particular regarding superficial slip and the tsunamigenic potential of the fault system.

The authors model the possible thermal surge amplitude and duration on the studied fault zone as a function of the seismic slip amount and of the chemical degradation kinetics, on several fault strands in the accretionary prism, seeking which ones show evidence of large seismic slip as a function of the inferred thermal surge.

Their interpretation is that at least 8 fault strands have indeed undergone large (in excess of 10 m) superficial seismic slip.

Methods and Rigour:

The detail of the chemical kinetics described here is beyond the reach of my knowledge, but it appears that the methodology is sound and well justified in thorough methods section and sup. Mat.

Attention is paid to the crucial error margin of the measurements and the uncertainty of the model, in a way that suggests rigour. Importantly, the measurements performed on core samples for the IODP borehole are not absolute measurements of the decomposition reaction, but relative to an undisturbed section of the same unit off-fault (other borehole).

This allows to mitigate the risk of misinterpreting chemical fluctuations within the geological sequence due to other causes than the co-seismic heating.

In regards to the seismic slip mechanicals and thermal diffusion model used to calculate the heating as a function of slip, the approach is fine, although I have a

few minor clarifications to ask (see below).

Recommendation:

I would recommend to publish this paper, after dealing with a few minor remarks below.

Questions, edits and possible clarifications:

The authors use the ratio r of chemical fraction between JFAST samples and Site436 borehole, which is shown in figure 2. I would also show a reference of variability (in the form of an error bar or std deviation) between the different analysed samples within the Site436 alone. This would allow to appraise the signal to noise ratio in r , and how much it differs from the random fluctuations within the geological section, with no further work and only a minor addition in the manuscript (possibly on Figure 2).

We agree that this is important and already reflected in the size of the box and whiskers in Figure 2. Because each point on Figure 2 represents one sample from the JFAST core, the size of the box and whisker gives a sense of the variability in the specific layer sampled. This is explained in the text in Lines 135-136, which we have edited for clarity.

Line 149. Eight faults show signatures compatible with large seismic slip, but how many faults have been analysed in total? It is not easy to infer the exact total number from the figures (although it may be indicated elsewhere in the text).

There are many faults within the JFAST core, but there are probably only three faults that we had samples near enough to detect a heat signature and did not (represented as gray faults on Figure 2). We have added this to line 157-159.

I could not find a mention of the possible time span of seismicity which is covered by this investigation. Is it possible to give an upper and lower bound ?

At the current convergence rate, the rocks we analyze have experienced at least 36,000 years of deformation (lines 325-327). We do not attempt any other time constraint.

Is each of the 8 “large slip” faults interpreted as an individual earthquake, or possibly there were fewer than 8 earthquakes but the rupture splayed on multiple faults, or is it not possible to discern?

The calculated slip in Figure 3 indicates the least slip possible that contributed to the heating anomaly on a given fault (meaning multiple earthquakes at that size adding up to the total displacement of 3.2 km could have yielded the observed biomarker anomaly). If only smaller earthquakes occurred, the heating anomaly

seen would not be as mature as it is (lines 264-266). This proxy does not indicate when the slip occurred, so it is not possible to distinguish whether slip occurred in individual events, multiple events, or whether multiple strands were slipping at the same time (lines 194-196).

Equation S1 re the thermal model. There is no factor of 2 in the original equation of cited work. Is it a matter of half-thickness versus whole thickness, or just a typo? Also the equation represents the evolution of temperature after slip cessation. Therefore it neglects the temperature rise phase integrated over the seismic slip duration in the chemical kinetics. This may be negligible, if the duration of the heated state (diffusion time) is large with respect to the slip duration. However, this probably should be indicated in the corresponding paragraph.

There was a missing 2 in our equation (now fixed). This was a typo, the original models were done correctly. This equation represents temperature both during and after the earthquake. We were missing a Heaviside function (again, this is the way our script works, we left it off in error), so that the second half of the equation is not considered during coseismic slip.

Minor edits:

The source in Table S3 for the co-seismic shear stress is not reference 16(Chester) but reference 23(Fulton).

Fixed

Figure 1 legend: replace "...regions of equal slip..." with "regions of equal slip as inferred from inversion of geophysical data..."

Done

Reference 22: add "USGS open file report 86508"

Done

With Best Regards,
Stefan Nielsen

Reviewer #3 (Remarks to the Author):

This is a most novel application of molecular thermal maturation indicators. In this study, a core (JFAST) taken in a region with multiple fault slips is compared to a nearby DSDP core where the faults are absent. The stratigraphy of the cores are closely correlated so that baseline values for the thermal parameters can be

obtained. The JFAST core has evidence of multiple seismic slippages as evidenced by localized frictional heating. The degree of unsaturation of alkenones and the carbon preference of plant derived n-alkanes are examined and the kinetic models, determined in prior studies, for these reactions are used to constrain the heating events during earthquake slippage.

This approach, to my knowledge, is unique and the influence of localized seismic heating on biomarkers has never been described. In general, the authors have tried to address the basic aspects of the problem and have tried to minimize likely error. The most obvious is establishing the starting (non-seismic) value from which the extent of reaction(s) are determined. I appreciate the care in which the authors have correlated the sediments between the DSDP and JFAST cores to determine these initial values. However, the CPI of immature sediments is so closely dependent on the influence of organic matter input that I'm not sure that the DSDP cores, regardless of elemental matches, accurately reflect the starting values. I suggest the authors add a paragraph or two that would explain the influence in the uncertainty of the initial values on the overall results.

We addressed the issue of whether the initial CPI values at Site 436 are relevant to the JFAST samples on lines 144-147. In that section we explained that the non-faulted *n*-alkane values at JFAST skew high compared to the range at Site 436. However, we explain that this would only mean that there could be more faults at JFAST that we have not identified as heated, given the conservative approach we take towards identifying something as anomalous compared to the population at Site 436. If we used non-faulted samples from JFAST for the initial *n*-alkane CPI values we would identify two additional heated faults in the JFAST core with CPI. However, those faults were already identified through alkenone anomalies, so our overall results would not change.

Values reported in Table S1 are being reported at a higher level of significance than is justified.

We have changed the reporting in Table S1 to use 3 significant figures in scientific notation. This accounts for the uncertainty in concentration values being fractional rather than absolute. As this is an excel spreadsheet we have changed the cell formatting to show this precision. We keep the full precision in the data themselves as any calculated sums and ratios could be in error.

All in all, an excellent study showing an new application of molecular markers, one certainly worthy of publication

Clifford Walters

Thank you!

REVIEWERS' COMMENTS:

Reviewer #2 (Remarks to the Author):

I think that the authors have addressed all my comments and the paper is ready to be published.

Regards,
Stefan Nielsen.

Reviewer #3 (Remarks to the Author):

I have verified that the revision has addressed the reviewers' concerns. I recommend that this paper be accepted for publication in Nature Communications.